# Human Brain Shows Recurrent Non-Canonical MicroRNA Editing Events Enriched for Seed Sequence with Possible Functional Consequence

**DOI:** 10.3390/ncrna6020021

**Published:** 2020-06-02

**Authors:** Deepanjan Paul, Asgar Hussain Ansari, Megha Lal, Arijit Mukhopadhyay

**Affiliations:** 1CSIR-Institute of Genomics & Integrative Biology, Delhi 110025, India; asgar.hussain@igib.in (A.H.A.); megha.lal@igib.in (M.L.); 2Academy of Scientific and Innovative Research, Ghaziabad, Uttar Pradesh 201002, India; 3Translational Medicine Unit, Biomedical Research Centre, University of Salford, Manchester M5 4WT, UK

**Keywords:** microRNA, RNA editing, human brain

## Abstract

RNA editing is a post-transcriptional modification, which can provide tissue-specific functions not encoded in DNA. Adenosine-to-inosine is the predominant editing event and, along with cytosine-to-uracil changes, constitutes canonical editing. The rest is non-canonical editing. In this study, we have analysed non-canonical editing of microRNAs in the human brain. We have performed massively parallel small RNA sequencing of frontal cortex (FC) and corpus callosum (CC) pairs from nine normal individuals (post-mortem). We found 113 and 90 unique non-canonical editing events in FC and CC samples, respectively. More than 70% of events were in the miRNA seed sequence—implicating an altered set of target mRNAs and possibly resulting in a functional consequence. Up to 15% of these events were recurring and found in at least three samples, also supporting the biological relevance of such variations. Two specific sequence variations, C-to-A and G-to-U, accounted for over 80% of non-canonical miRNA editing events—and revealed preferred sequence motifs. Our study is one of the first reporting non-canonical editing in miRNAs in the human brain. Our results implicate miRNA non-canonical editing as one of the contributing factors towards transcriptomic diversity in the human brain.

## 1. Introduction

Biological systems thrive on diversity. At the molecular level, diversity is often contributed by variations occurring at different steps of information processing, namely DNA to RNA and RNA to protein. RNA sequences can differ from their DNA templates owing to transcriptional errors and RNA editing [1,2]. Proofreading and repair mechanisms of transcription keep a check on transcriptional errors [3,4], making RNA editing one of the major contributors to RNA sequence diversity. The importance of RNA editing in diversity is even higher in tissues or organs where cells are not actively dividing—such as post-mitotic neuronal cells [5,6,7]. The predominant RNA editing event, mediated by ADAR (Adenosine Deaminase acting on RNA) results in adenosine-to-inosine change (A-to-I) in RNA [8,9,10,11,12,13]. Along with A-to-I, enzyme-mediated editing also includes cytosine-to-uracil change (C-to-U) mediated by APOBEC (Apolipoprotein B mRNA-editing enzyme, catalytic polypeptide-like) [14,15]. A-to-I and C-to-U are categorized as canonical RNA editing. Initially, editing was identified in mRNAs. However, recent reports find the majority of the editing events within the non-coding regions of the genome [9,16,17,18]. Within non-coding RNAs, a large proportion of editing is found in Alu elements [19,20]. Elevated Alu editing is reported in multiple cancers leading to an increase in transcriptomic diversity [21]. Recent advances in next-generation sequencing technologies have enabled researchers to investigate non-canonical editing events (other than A-to-I and C-to-U). The mechanisms behind these events are unknown and broadly classified as RNA–DNA differences (RDD). A recent study identified 10,000 exonic sites differing between RNA and DNA in B-cells, which translated into variant protein sequences implicating their functional relevance [22]. A few groups suggested these RDDs could be errors due to mapping, sequencing and genetic variations [23,24,25] However, the authors of the original study reproduced RDDs using orthogonal methods for both experiment and analysis [26]. Another independent study also found RDDs in human cytokine receptor, IL12RB1 [27]. IL12RB1 is important for human resistance against pathogens [28], and variation in IL12RB1 has been associated with multiple disease susceptibility [29]. It will be interesting to see whether RDDs in IL12RB1 can modify intra-individual and inter-tissue differences towards disease susceptibility. Further, a study also identified RDDs in proto-oncogenes and tumour suppressors [30]. Apart from these, several other studies have identified RDDs in different tissues [31,32,33]. These evidences show that, although the mechanism and extent of non-canonical editing is unknown, these are indeed true events with likely functional consequences and need further research. 

MiRNAs usually binds to the 3′-untranslated region (3′-UTR) of mRNAs leading to downregulation of gene expression [34]. The target recognition is facilitated by a protein complex called RISC which loads the mature miRNA and lodges it onto the 3′-UTR of the target mRNA to direct post-transcriptional repression [35,36]. In animals/metazoans, extensive complementarity between miRNA and its target leads to translational suppression and mRNA destabilization and degradation [37,38,39]. In particular, bases 2–8 from the 5′-end of the mature miRNA constitute the seed sequence and are critical for target recognition [40]. A-to-I editing within microRNAs has been well studied and can affect its processing [41,42] and result in redirection of targets [43]. MiRNA editing is implicated in tissue-specific regulation in both health and disease [41,42,43,44,45,46,47,48,49,50]. Reports of non-canonical editing in microRNAs are scarce [51,52,53] and the validity as well as functionality of these events needs further study. In this study, we have used massively parallel next-generation sequencing and analysed non-canonical editing in mature microRNAs in two different regions of normal human brain—namely, the frontal cortex and corpus callosum.

## 2. Results and Discussion

We have performed small RNA sequencing in paired samples of frontal cortex (FC) and corpus callosum (CC) from nine individuals (post-mortem samples) to identify editing events in the mature microRNA population (Appendix A).

We overlapped the identified miRNA variant sites with dbSNP version151 (All SNPs) followed by exome data to filter out variations at the DNA level. However, it is worthwhile to note here that these variations (genome encoded) can still be in the resulting transcript (miRNA) and bring out possible functional consequences [54,55]. Further, all A-to-C modifications were excluded from the analysis due to specific biases of Illumina sequencers [56,57].

After filtering out these variations at the DNA level, the total number of canonical edits identified in FC were 255 (A-to-I = 211 and C-to-U = 44) and in CC were 177 (A-to-I = 136 and C-to-U = 41). In case of non-canonical miRNA editing, 175 and 158 events were detected in FC and CC, respectively (Appendix A). This observation was in agreement with previous literature reporting a higher proportion of canonical editing, especially A-to-I, within mature miRNAs [51,52]. Interestingly, we observed inter-individual and intra-individual differences in terms of relative proportions of canonical and non-canonical editing events in mature miRNAs (Figure 1, Appendix A). For example, we detected non-canonical editing events viz., miR-454_U-to-G, miR-30e_U-to-C etc., which were enriched in either FC or CC only—but not in both—indicating intra-individual tissue-specific non-canonical events (Table 1).

Out of total non-canonical events, 113 and 90 events were unique in FC and CC, respectively (Figure 1A,B). More than 15% of these events recurred in at least three samples in both FC and CC (Figure 1A,B and Appendix A). While variation at any position within a microRNA can have an impact [41,46], a seed sequence variation can lead to target redirection [36]. Thus, a variation in the seed sequence can have a larger functional impact. Interestingly, 71.68% (81 out of 113) and 74.44% (67 out of 90) of the non-canonical editing events were within the seed sequence in FC and CC, respectively (Figure 1A,B). The overlap between the predicted mRNA targets of the unedited and edited miRNAs was less than 10% (Table 1 and Appendix A), confirming the possible target redirection. The pathways enriched for the unedited and edited miRNAs targets were different. For example, gene ontology and pathway enrichment analysis for the unedited miR-433 revealed genes involved in neuron development and differentiation, and pathways relating to chemical synapses and neuronal systems (Appendix A). Additionally, the edited version of this miRNA resulted in gene sets related to cell motility and locomotion (Appendix A). It is noteworthy that miR-433 showed considerable levels of editing (up to 36.6%) and was identified in six samples (FC and CC combined) which indicates an important biological role of the edited miRNA. It will be interesting to pursue if the unedited and the edited version both remains relevant in separate distinct stages of neurogenesis (e.g., axonal growth vis-à-vis pruning).

Within non-canonical editing events found in our study, C-to-A and G-to-U events accounted for over 80% (Figure 2). C-to-A events were most abundant (44.24% (50/113) in FC and 54.44% (49/90) in CC), followed by G-to-U events (39.82% (45/113) in FC and 34.44% (31/90) in CC) (Figure 2A,B). Additionally, over 70% of the C-to-A and G-to-U events were within the seed sequence (Figure 2).

We have analysed whether sequence contexts in the miRNAs accounted for the two abundant events mentioned above. For both types, we analysed upstream and downstream sequence context with the edited base in the middle. For C-to-A-edited miRNAs, there was a preference for adenosine (*p* = 0.02) downstream of the edited cytosine and avoidance for guanosine on both sides (*p* = 0.02 and *p* = 0.009). We also found a lack of uracil (*p* = 0.04) downstream of the edited cytosine (Figure 3 and Appendix A). Additionally, “UCA” (13/58), “ACC” (13/58) and “CCA” (7/58) accounted for more than 55% of the triplets present in miRNAs harbouring C-to-A editing. Analysis of 1871 pre-miRNAs revealed that the fractions of abovementioned triplets in pre-miRNAs were 0.017, 0.012 and 0.018, respectively (Appendix A). For the G-to-U-edited miRNAs, a preference for adenosine both upstream (*p* = 0.002) and downstream of the edited guanosine was observed (Figure 3A and Appendix A). “AGG” and “AGA” were the top two enriched triplets (0.19 and 0.13, respectively) observed for miRNAs harbouring G-to-U editing. The corresponding frequencies of these triplets in in pre-miRNAs were 0.023 and 0.018, respectively (Appendix A).

For the two most abundant editing types discussed above, the observed enrichment of motifs in editing events is not due to a preferred DNA sequence context common to miRNAs (Appendix A). It is therefore likely that such enrichments indicate distinct, yet unknown, molecular mechanisms that recognize and utilize these motifs for post-transcriptional modifications. The non-canonical events we reported can be considered as isomiRs under the category of nucleotide substitution events, which encompass a huge range of changes including A-to-I modifications [58,59,60]. Further studies and validation of non-canonical editing by orthogonal methods (Sanger sequencing) will be necessary to shed more light on such possible mechanisms.

Interestingly, a recent genome-wide study detected non-canonical miRNA editing events and found that most of the U-to-G changes were detected in the superior frontal gyrus of brain samples [53]. We have also detected ten unique U-to-G editing events in our FC and CC samples. Additionally, the above study identified G-to-U seed editing events within let-7 members (also found in our study, Table 1) leading to specific enriched pathways. Enrichment of specific non-canonical editing in specific regions of the brain can have important biological consequences and needs further study. Single-cell sequencing of different brain cells will provide a better understanding of the functional role of non-canonical editing. We have shown that non-canonical editing in miRNA, especially in the seed sequence, is a likely contributor to transcript diversity with potential functional consequences in the human brain.

## 3. Materials and Methods

### 3.1. Samples

Portions of frontal cortex and corpus callosum were obtained from the NIMHANS Brain Bank, Bengaluru, India. These were collected post-mortem from road accident victims. The samples were ethically cleared for research purposes through the brain bank. Additionally, the institutional review board has provided ethical approval for the study. The study also adhered to the Declaration of Helsinki for the use of human samples. The details of the samples used in the study are provided in Appendix A.

### 3.2. RNA Isolation, Library Preparation and Sequencing

Briefly, total RNA was isolated from frontal cortex and corpus callosum using a miRvana miRNA isolation kit (Ambion, Austin, TX, USA) as per the manufacturer’s instructions. Libraries were prepared using Illumina’s TruSeq Small RNA Sample Prep Kit following the manufacture’s protocol. Two samples were loaded in one lane and 50-base single-end sequencing was done on an Illumina HiSeq 2000 platform. The small RNA sequencing data can be accessed in the sequence read archive (SRA ID: SRP063390).

### 3.3. Data Analysis

Data analysis was done using the published pipeline [61] with the default parameters. Briefly, low-quality reads and adapters were removed. Filtered reads were aligned to the genome using Bowtie 0.12.7, allowing unique best alignment and one mismatch. Next, reads were mapped to pre-miRNA sequences. Binomial statistics were used to remove sequencing errors. Finally, Bonferroni correction was used to identify statistically significant variations. dbSNP version 151 (All SNPs) and exome sequencing data were used to filter out variations at the DNA level. The raw output of the miRNA editing pipeline is provided in Appendix A.

### 3.4. Target Prediction and Functional Enrichment Analysis

Target prediction for unedited and edited miRNAs was done using isoTar version 1.2 [62]. We selected the mRNA targets predicted by four out of five independent miRNA target prediction tools; viz. PITA, RNAhybrid, TargetScan, miRanda and miRmap. Following that, we did gene ontology and REACTOME pathway enrichment analysis using gene set enrichment analysis [63,64].

### 3.5. Sequence Context for the Edited miRNAs

Unique non-canonical miRNA editing events combining FC and CC were used for analysis. This led to 58 C-to-A and 48 G-to-U events. One base upstream and one downstream of the edited base were fetched using an in-house script for the abovementioned edited miRNAs. Sequence preference around the edited base was visualized using WebLogo 3 [65]. To look for tri-nucleotide (triplet) sequence composition within pre-miRNAs, human pre-miRNAs were downloaded from miRBase version 20 and a three-base sliding window was used to analyse enriched tri-nucleotides using in-house scripts. The entire statistical test relating to calculation of sequence context around the edited nucleotide has been done using a two-tailed test for proportion.

The methods followed in this paper can be seen in more detail in our previous publication [47].

## Figures and Tables

**Figure 1 ncrna-06-00021-f001:**
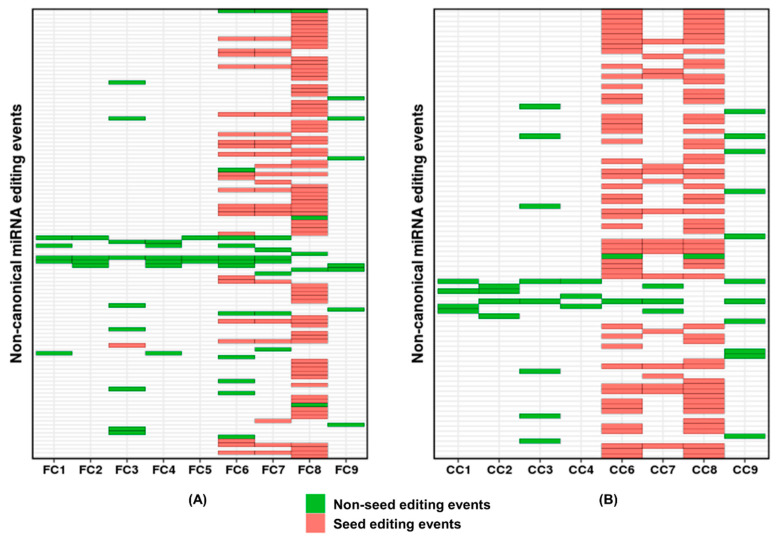
Distribution of non-canonical editing within mature miRNAs in FC and CC. The figure shows 113 and 90 non-redundant non-canonical miRNA editing events and their overlap across nine (**A**) FC and (**B**) CC, respectively. Each row represents a single miRNA editing event. The events are alphabetically arranged according to miRNA name. For details, refer to Appendix A. Seed editing events are represented in pink and the non-seed in green. We did not detect any non-canonical miRNA editing events in CC5.

**Figure 2 ncrna-06-00021-f002:**
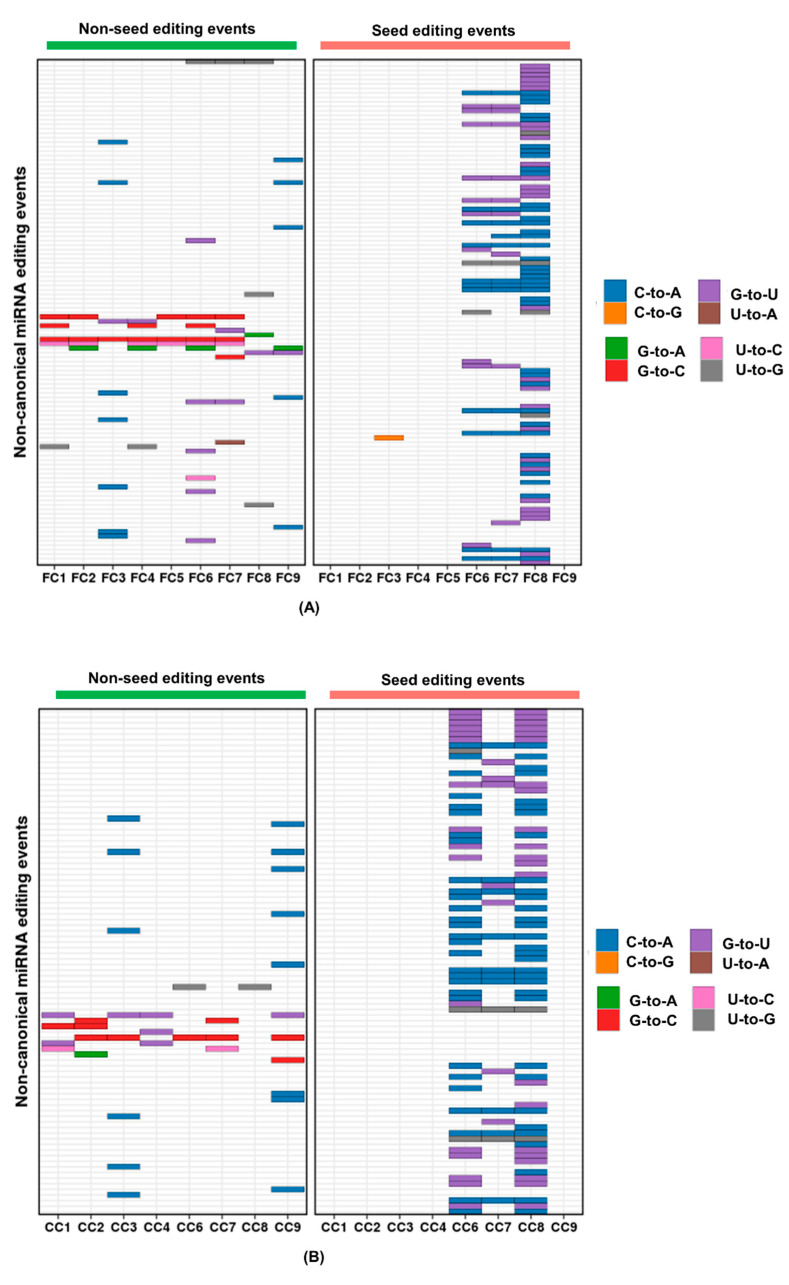
Distribution of different non-canonical editing within mature miRNAs. The different colours represent the types of non-canonical editing and their distribution within mature miRNA detected in (**A**) FC and (**B**) CC. The pink and the green bar over the panels correspond to seed and non-seed editing events from Figure 1. The events are alphabetically arranged according to miRNA name. For details, refer to Appendix A. A-to-U modification was not detected in the study. C-to-G and U-to-A modifications were not detected specifically in CC.

**Figure 3 ncrna-06-00021-f003:**
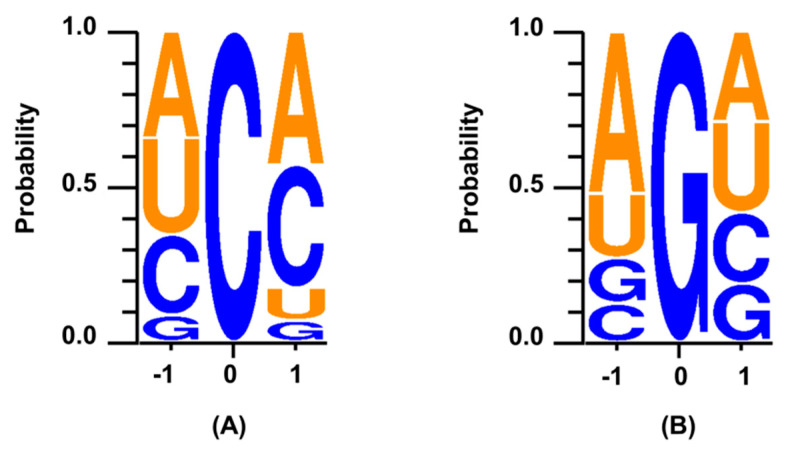
Analysis of sequence context for the non-canonical editing events. The figure shows the edited base in the middle (position 0) and the probability of the bases one base upstream (position −1) and downstream (position +1). (**A**) For C-to-A edited miRNAs, adenosine was a preferred base downstream (*p* = 0.02) and guanosine was avoided on either side of the edited cytosine (*p* = 0.02 and *p* = 0.009). Uracil was also not preferred downstream to the edited cytosine (*p* = 0.04). (**B**) For G-to-U-edited miRNAs, adenosine was the preferred base both upstream (*p* = 0.002) and downstream of the edited guanosine.

**Table 1 ncrna-06-00021-t001:** Target prediction analysis of non-canonical miRNA editing events present in at least three FC and CC samples.

MiRNA	Strand/Position	Seed	Editing Types	Presence in Samples (FC/CC)	^a^ Target Prediction (Before/After Editing)	Overlap (%)
hsa-miR-100	5p/3	Yes	C-to-A	3/3	30/16	0 (0)
hsa-miR-127	3p/4	Yes	G-to-U	3/3	128/60	0 (0)
hsa-miR-146b	5p/4	Yes	G-to-U	3/2	502/111	9 (1.79)
hsa-miR-181c	5p/3	Yes	C-to-A	3/3	207/72	1 (0.48)
hsa-miR-181d	5p/3	Yes	C-to-A	3/2	580/269	21 (3.62)
hsa-miR-204	5p/3	Yes	C-to-A	3/3	498/90	6 (1.20)
hsa-miR-221	3p/4	Yes	U-to-G	3/0	378/103	5 (1.32)
hsa-miR-23a	3p/3	Yes	C-to-A	3/3	258/73	2 (0.78)
hsa-miR-23b	3p/3	Yes	C-to-A	3/3	372/106	5 (1.34)
hsa-miR-26b	5p/3	Yes	C-to-A	3/3	93/41	0 (0)
hsa-miR-301a	3p/4	Yes	U-to-G	2/3	184/508	7 (3.80)
hsa-miR-421	Mature/3	Yes	C-to-A	3/3	248/122	5 (2.02)
hsa-miR-433	3p/3	Yes	C-to-A	3/3	403/166	10 (2.48)
hsa-miR-99a	5p/3	Yes	C-to-A	3/3	35/23	0 (0)
hsa-miR-99b	5p/3	Yes	C-to-A	3/2	49/22	0 (0)
hsa-miR-454	3p/4	Yes	U-to-G	0/3	232/622	21 (9.05)
hsa-let-7a-1	5p/9	No	U-to-G	3/0	NA	NA
hsa-miR-30a	5p/19	No	G-to-C	5/2	NA	NA
hsa-miR-30a	5p/19	No	G-to-U	2/4	NA	NA
hsa-miR-30d	5p/19	No	G-to-C	3/2	NA	NA
hsa-miR-30e	5p/19	No	G-to-C	7/5	NA	NA
hsa-miR-30e	5p/17	No	U-to-C	6/2	NA	NA
hsa-miR-30e	5p/18	No	G-to-A	4/1	NA	NA

^a^ Percentage overlap was calculated by overlapped targets/targets before editing. NA—Not applicable because they were not seed editing events.

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
