# Peer review of "Human Brain Shows Recurrent Non-Canonical MicroRNA Editing Events Enriched for Seed Sequence with Possible Functional Consequence"

_ncrna, 2020, doi:10.3390/ncrna6020021_

Round 1
Reviewer 1 Report
In the revised version of the manuscript, Deepanjan Paul et al. have well answered all my comments/questions and took all the suggestions I gave them. The current version article is significatively improved, but before being considered for publication, I have a couple comments/questions, listed below, that need to be addressed.
Majors:
- In the Introduction section, although it provides now sufficient background and includes relevant references, the authors should better structure it and add a few more details about miRNAs and miRNA targeting. Moreover, the authors should avoid redundancy, as is the case for: “The predominant RNA editing event, mediated by ADAR (Adenosine Deaminase acting on RNA) results in adenosine to inosine change (A-to-I) on RNA” and “A-to-I editing is the most well studied and predominant editing event in both coding and non-coding transripts.” Also, note that wording in the last word of the latter sentence is wrong.
- Results provided by the authors are quite innovative and might meet with skepticism in the scientific community. Despite the overlap of some non-canonical editing events with those reported by Zheng et al., 2016, it would be appropriate to verify even a small random subset (e.g., six) of non-canonical editing events for each major non-canonical editing type – in this case C-to-A and G-to-U. This validation can be accomplished by performing Sanger sequencing according to examples found in Kawahara et al., 2008 and Li et al., 2018. Noteworthy, in the latter study, the application of Sanger sequencing showed that the C-to-U editing events found in pri-miRNAs by HTS data analyses were a mere artifact of the HTS procedure. In light of this, does Sanger sequencing fully confirm the results reported in the present study? In the event the authors are unable to perform these experiments (i.e., Sanger sequencing), they have to state (in the conclusion part) the necessity for other groups to do them to validate the results.
- The authors employed isoTar tool for the prediction analyses, as I suggested. I have a question about that: recently, the developers of isoTar released a new version of this tool (v.1.2), correcting some parameters of miRanda predictor. In light of this, did the authors use the latest version of isoTar for the prediction analyses? If not, I strongly suggest updating the prediction results by using the 1.2 version of isoTar and consequently updating the enrichment analyses results.
Minors:
- The wording “(FC/CC)” should be added in column six of Table 1, after “Target prediction.”
Reviewer 2 Report
Authors adequately addressed all concerns.
Author Response
We thank the reviewer for his/her comments. It has certainly made the manuscript better.
Round 2
Reviewer 1 Report
I have just a few minors that need to be addressed before being accepted. I listed them below:
- Abstract section:
- "Cytosine-to-Uracil changes constitutes canonical editing. Rest is non-canonical editing." Replace with "The rest is" or "The remaining is."
- Introduction section:
- "[14,15]; together representing canonical RNA editing." Replace the semicolon after references 14-15 with a dot and correctly rewrite the following sentence.
- "Apart from mRNAs and Alu sequences," Please, remove this sentence.
- "Extensive complementarity between miRNA and its target leads to mRNA cleavage... " Rewrite as: "In animals/metazoans, extensive complementarity between miRNA and its target leads to translational suppression, and mRNA destabilization and degradation. In particular, bases 2-8 from the 5’-end of the mature miRNA constitute the seed sequence and... ".
Author Response
Response to Reviewer's Comments
1. Abstract section:Cytosine-to-Uracil changes constitutes canonical editing. Rest is non-canonical editing." Replace with "The rest is" or "The remaining is."
Response 1: It has now been changed to "The rest is non-canonical editing."
2. Introduction section:"[14,15]; together representing canonical RNA editing." Replace the semicolon after references 14-15 with a dot and correctly rewrite the following sentence.
Response 2: The semicolon has been replaced with a dot. The sentence has been rewritten as "A-to-I and C-to-U are categorized as canonical RNA editing" in the revised manuscript.
3. "Apart from mRNAs and Alu sequences," Please, remove this sentence.
Response 3: This sentence has been removed. The paragraph starts with "MiRNAs usually binds to 3’-untranslated region (3’-UTR) of mRNAs leading to downregulation of gene expression".
4. "Extensive complementarity between miRNA and its target leads to mRNA cleavage... " Rewrite as: "In animals/metazoans, extensive complementarity between miRNA and its target leads to translational suppression, and mRNA destabilization and degradation. In particular, bases 2-8 from the 5’-end of the mature miRNA constitute the seed sequence and...".
Response 4: This is rewritten as suggested "In animals/metazoans, extensive complementarity between miRNA and its target leads to translational suppression, and mRNA destabilization and degradation". For this part an additional reference has been included in the revised version. This is followed by the addition of "In particular, bases 2-8 from the 5’-end of the mature miRNA constitute the seed sequence and...".
We thank the reviewer for the detailed review of our manuscript. It has certainly enhanced the quality of the manuscript.
This manuscript is a resubmission of an earlier submission. The following is a list of the peer review reports and author responses from that submission.
Round 1
Reviewer 1 Report
In this Brief Report, authors investigated the abundance of non-canonical RNA editing of microRNAs (miRNA) in 10 samples collected from the frontal cortex and corpus callosum of human brains. Based on presented results, authors associated non-canonical editing of miRNA with the transcriptomic diversity in the human brain. Overall, experimental design was elegantly planned and executed. In addition, main conclusions are supported by the data presented. However, I do have some minor concerns that would improve this otherwise exciting manuscript:
1) Introduction: Sentence stating “…thereby making RNA editing a major contributor towards RNA sequence diversity. may be seen as too strong of a statement. Authors may want to say that RNA editing is one of the major contributors to sequence diversity.
2) Authors could explain in the introduction importance of the seed sequence since it is relevant for results.
3) Authors may want to clarify if same portions of the frontal cortex were used to avoid possible inter-regional differences. Best would be if authors can clearly mention which are was used for the analysis
4) This manuscript would benefit from inclusion of human brain ages used in the study. Also, authors should report post-mortem delay for each of the 10 samples used.
5) Authors may want to cite following study: “1: Li L, Song Y, Shi X, Liu J, Xiong S, Chen W, Fu Q, Huang Z, Gu N, Zhang R. The landscape of miRNA editing in animals and its impact on miRNA biogenesis and targeting. Genome Res. 2018 Jan;28(1):132-143.”
6) Authors may want to state more clearly how much does this study overlap with their previous report published in 2017, Paul et al. in Scientific Reports.
7) Results: 114 out of 182 is 62.63%, not what authors report 62.3%
8) Figure 1, 2: In the legend, authors need to state that they are presenting non-canonical events in miRNAs. Authors may want to make a legend bar denoting in the image what does each color/tone present.
9) Figure 1: Although reader can assume, authors may want to state clearly that light shades of green are non-canonical events in the miRNA seed sequence.
Reviewer 2 Report
In this paper, Paul and Mukhopadhyay have analyzed small RNA-seq data from the frontal cortex (FC) and corpus callosum (CC) from 10 normal individuals aim at investigating non-canonical post-transcriptional RNA modifications in microRNA molecules. In this study, the authors discovered hundreds of unique non-canonical post-transcriptional RNA modifications in miRNAs. Despite the results presented by the authors are of general interest to the community, I have several concerns as follows:
Major concerns:
In the introductive section, the authors briefly summarized concepts concerning RNA editing, distinguishing between canonical and non-canonical editing events. However, it should be also reported that according to the results from recently published high impact factors papers (add appropriate references), A-to-I editing occurs at much higher rates than the other editing types (including the C-to-U), thus being by far the most representative one. Even a brief description of miRNAs and their function is necessary for a better understanding of the topic. In the part: “A recent study identified 10,000 exonic sites differing between RNA and DNA in B-cells, which were also detected at the protein level implicating their functional relevance [24]”, the authors should also cite and argument other three papers as comments of the one they mention here: (1) Lin et al. Science (2012), (2) Kleinman et al. Science (2012) and (3) Pickrell et al. Science (2012). Before comparing frequencies of non-canonical editing types with each other, it would be very important to compare total levels of canonical (1) A-to-I and (2) C-to-U editing with total levels of (3) non-canonical editing events on the same set of retrieved miRNAs. An example is given in Picardi et al., 2015, Scientific Reports, where frequencies of potential non-canonical events were shown and compared with the canonical ones. Are the results obtained by the authors comparable with those obtained by previous studies? Also, results obtained for A-to-I and C-to-U should be reported on a distinct supplementary table, and the number of edited reads/total reads should be shown for each site-specific editing event, as it was done in Supplementary Table 1. It also would be very useful to create a Supplementary Table with the output of the pipeline the authors used for the miRNA editing detection. Regarding the pipeline used for miRNA editing detection, the authors did not mention which criteria of filtering of the significance they used (together with the correction: BH or Bonferroni). Also, the dbSNP build (142 considered in this study is very dated (of 2014), when now there is available the build 151 in Human, because of that strongly suggest updating the version of dbSNP in order to filter out any new DNA modification. To execute the target prediction analysis, the authors used TargetScanHuman 5.2 Custom. However, the usage of a consensus of miRNA predictor tools would allow more strong predictions. In this case, I would recommend taking into consideration for this paper to apply isoTar developed by Distefano et al., a consensus miR target predicted tool able to perform prediction analyses for canonical and modified miRNAs. As a consequence of the previous comment, pathway analyses should be carried out for results provided by the suggested target prediction tools. No information is provided by the authors concerning the statistical test applied to determine the significance of the three-nucleotide motifs identified by WebLogo. At the end of the “Results and Discussion” section, the authors stated that “non-canonical editing can have major consequences on miRNA function, particularly for those in the seed sequence indicating functional relevance in creating diversity, especially in neuronal regions of brain with limited regeneration potential.” The problem here is that non-canonical editing can have major consequences on miRNA function respect to what? If the authors meant respect to the “canonical miRNA editing phenomenon”, they also need to show results to support this claim.Minors:
Broad revising of English grammar and syntax is required. In the sentence “Importance of RNA editing in diversity is even higher in tissues or organs where cells are not actively dividing – such as post mitotic neuronal cells” the authors have to add a citation. In the sentence “This agrees with previously report of seed sequence A-to-I editing events in mature miRNAs in various human tissues including brain [20].”, the authors just cited their previous work (Paul D. et al., Scientific Reports 2017). Please, cite also: Li et al., Genome Research 2018, Zhang et al., Nucleic Acid Research 2016 (Ref. 28 in the present manuscript) and Picardi et al., Scientific Reports 2015. In “a seed sequence variations directly implicates a redirection of target mRNA as a functional consequence of the mature miRNA.”, please cite Bartel, Cell 2009. In “Interestingly, we did not detect any A-to-U changes in either FC or CC samples. A-to-C modifications were excluded from the analysis due to specific biases of illumine sequencers (Nakamura et al., 2011).” the citation is missing. Also, is “illumine” means “Illumina”. In “The target prediction for unedited and edited miRNAs was done using TargetScanHuman 5.2 Custom [30,31].” The citation “31” is wrong. The authors should mention the version of miRBase int the materials and methods sections.